# Cascade Screening and Treatment Initiation in Young Adults with Heterozygous Familial Hypercholesterolemia

**DOI:** 10.3390/jcm10143090

**Published:** 2021-07-13

**Authors:** Amy L. Peterson, Matthew Bang, Robert C. Block, Nathan D. Wong, Dean G. Karalis

**Affiliations:** 1Department of Pediatrics, Division of Pediatric Cardiology, University of Wisconsin School of Medicine and Public Health, Madison, WI 53792, USA; 2Heart Disease Prevention Program, Division of Cardiology, School of Medicine, University of California Irvine, Irvine, CA 92697, USA; mrbang@uci.edu (M.B.); ndwong@hs.uci.edu (N.D.W.); 3Department of Public Health Sciences and Cardiology Division, Department of Medicine, University of Rochester Medical Center, Rochester, NY 14642, USA; robert_block@urmc.rochester.edu; 4Department of Cardiology, Thomas Jefferson University Hospital, Philadelphia, PA 19107, USA; dean.karalis@jefferson.edu

**Keywords:** familial hypercholesterolemia, dyslipidemia, atherosclerosis, cascade screening, primary prevention, cholesterol screening

## Abstract

Heterozygous familial hypercholesterolemia (HeFH) creates elevated low-density lipoprotein cholesterol (LDL-C), causing premature atherosclerotic cardiovascular disease (ASCVD). Guidelines recommend cascade screening relatives and starting statin therapy at 8–10 years old, but adherence to these recommendations is low. Our purpose was to measure self-reported physician practices for cascade screening and treatment initiation for HeFH using a survey of 500 primary care physicians and 500 cardiologists: 54% “always” cascade screen relatives of an individual with FH, but 68% would screen individuals with “strong family history of high cholesterol or premature ASCVD”, and 74% would screen a child of a patient with HeFH. The most likely age respondents would start statins was 18–29 years, with few willing to prescribe to a pediatric male (17%) or female (14%). Physicians who reported previously diagnosing a patient with HeFH were more likely to prescribe to a pediatric patient with HeFH, either male (OR = 1.34, 95% CI = 0.99–1.81) or female (OR = 1.31, 95% CI = 0.99–1.72). Many physicians do not cascade screen and are less likely to screen individuals with family history of known HeFH compared to “high cholesterol or premature ASCVD”. Most expressed willingness to screen pediatric patients, but few would start treatment at recommended ages. Further education is needed to improve diagnosis and treatment of HeFH.

## 1. Introduction

Familial hypercholesterolemia (FH) is an autosomal dominant disorder characterized by markedly elevated levels of low-density lipoprotein cholesterol (LDL-C) that are present from birth and predispose affected individuals to premature onset of atherosclerotic cardiovascular disease (ASCVD). In its heterozygous form (HeFH), it affects 1 in 200–300 individuals [1,2,3,4,5], making it the most common monogenic disorder and the most common potentially fatal genetic disease in humans. Early treatment of FH correlates directly with prevention of ASCVD and death [6,7,8]. Current HeFH treatment guidelines from the National Lipid Association, the National Heart, Lung, and Blood Institute with the American Academy of Pediatrics [9], and the American Heart Association [10] recommend initiation of medical therapy with a hydroxymethylglutaryl-CoA reductase inhibitor (“statin”) starting as young as age 8. 

HeFH screening recommendations are more controversial. Cascade screening children from families with HeFH as young as 2 years of age is recommended by all current HeFH guidelines [11,12,13]. However, universal cholesterol screening of all children to detect HeFH is recommended by the American Academy of Pediatrics [9], but was given a level of evidence “I” (insufficient evidence to evaluate harms and benefits) from the United States Preventive Services Task Force [14]. The 2018 Multi-Society guidelines for cholesterol management assigned universal pediatric cholesterol screening a 2B recommendation, implying the intervention “may be reasonable” and “effectiveness is unclear” [15]. Universal pediatric cholesterol screening has been shown to increase detection of HeFH [16] but these guidelines are not well-recognized by physicians [17,18,19], which may lead to under-recognition of HeFH in children and young adults.

In adults, studies evaluating clinician screening and treatment of hypercholesterolemia have also yielded suboptimal results. In the US, 60% of young adults with a personal history of high cholesterol reported having a cholesterol panel checked within the previous 5 years, and only 15% of 20–39-year-olds with severe hypercholesterolemia had been prescribed a statin [20]. When presented with a case example, fewer than 30% of cardiologists correctly diagnosed HeFH [21]. The aim of our study was to measure physicians’ self-reported compliance with recommendations and knowledge regarding current evidence supporting cascade screening of relatives potentially affected by HeFH, and for timing of statin initiation in young adults and children. Although most cardiologists and many primary care physicians (PCPs) do not directly treat pediatric patients, it is important for them to be familiar with guidelines so they are able to recommend that children of their adult patients with HeFH seek cholesterol screening from the child’s physician. We hypothesized that the ability of cardiologists and PCPs to diagnose, treat, and cascade screen patients with HeFH was poor.

## 2. Materials and Methods

The National Lipid Association surveyed 500 cardiologists and 500 PCPs (consisting of family medicine, general medicine, and internal medicine specialties). The survey was conducted from 29 August to 30 September 2019. Inclusion criteria included: currently licensed and practicing within the United States, and self-report of having evaluated at least one patient with baseline LDL-C ≥ 190 mg/dL. Physicians who reported certification by the American Board of Clinical Lipidology or Accreditation Council for Clinical Lipidology were excluded, as were physicians with a primary specialty of general pediatrics or a pediatric subspecialty.

The survey was conducted online by MedSurvey (Southampton, PA, USA). Physicians meeting eligibility criteria could complete the survey, and once the goal enrollment of 1000 physicians (500 PCPs and 500 cardiologists) was reached, the survey was closed. If physicians met eligibility criteria based on a series of initial questions, they were invited to continue with the survey and received a small honorarium for completing it. The eligibility questions aimed to minimize geographical area bias within the United States. The study used deidentified data and was deemed exempt from Institutional Board Review.

The survey consisted of 30 questions assessing physician demographics and recognition and treatment of HeFH (Appendix A). All physicians self-reported whether or not they routinely cared for individuals under 18 years of age. Likert scales were used to measure the likelihood of screening first-degree family members and age at which cholesterol screening and statin treatment in patients with HeFH should begin. For respondents indicating that they would not perform cholesterol screening in individuals under 18 years, explanations were sought.

Survey data were collected and analyzed by the independent research partner, with supplemental analysis by the authors. Categorical variables were compared between PCPs and cardiologists using Chi-square test of proportions. Multiple logistic regression with stepwise selection was also performed to examine whether provider type, region, practice setting, or previous diagnosis of a patient with HeFH were independently associated with intention to start a patient with HeFH under the age of 18 on a statin. This was analyzed for all pediatric patients and separated into male and female patients under 18 years old. A *p*-value < 0.15 was chosen for entry in the stepwise regression. When a final regression model of predictors was constructed, those of borderline significance (0.05 < *p* < 0.15) still contributed significantly to variance and precision of the final model, which provided greater precision of estimates, including those that were significant. Predictors with *p* > 0.15 were excluded from the model. Odds ratios (ORs) and 95% confidence intervals (CI) were determined.

## 3. Results

A total of 1561 physicians responded to the invitation to complete the survey, with 561 failing to meet eligibility criteria. A total of 1000 physicians completed the online survey. This included 500 cardiologists and 500 PCPs. Among PCPs, 46% identified as a family medicine physician, 46% as an internal medicine physician, and 8% as a general practitioner. Of all physicians, 53% reported practicing in a suburban location, 37% in an urban location, and only 10% in a rural location. For practice setting, 57% reported private practice, followed by 24% employed by a health system, 17% in a primarily academic setting, and 1% each in the Veterans Affairs/government health system or other. Regional differences in practice location were minimal, with physicians reporting they practiced in the Northeast (22.3%), Southeast (21.4%), Midwest (20.0%), Pacific (19.4%), or Southwest (16.9%) of the United States. Of cardiologists 17% and of PCPs 58% reported treating patients under 18 years old for any medical condition. 

Of PCPs 67% and of cardiologists 69% reported that they monitor LDL-C levels in an individual with a strong family history of high cholesterol or premature ASCVD starting between 18 and 29 years of age (*p* = 0.86) (Figure 1). For a patient with known HeFH, 54% (58% of cardiologists, 50% of PCPs, *p* = 0.94) indicated that they would “always” recommend cascade screening in an adult first-degree relative (Figure 2). Mean Likert score for the entire cohort was 4.3.

Seventy-four percent of all respondents reported that they would recommend checking cholesterol in a child of a patient with known HeFH, with physicians who reported caring for pediatric patients more likely than those who do not to report screening in childhood (86.9% vs. 65.9%, *p* < 0.001). However, only 7.3% of all respondents reported that they would recommend screening children between 2 and 8 years of age, with respondents caring for pediatric patients more likely to screen at this age (8.0% vs. 6.9%, *p* < 0.001) (Figure 3).

Among the 262 respondents who would not screen until adulthood, 49 (18.7%) reported that they cared for pediatric patients. For all 262 respondents, the most common reason cited was lack of familiarity with pediatric cholesterol guidelines (75.6%). Respondents who reported that they care for pediatric patients were less likely than those who do not care for pediatric patients to report lack of familiarity with pediatric cholesterol guidelines, but were more likely to agree with the statements that “there is insufficient evidence that screening children prevents ASCVD” and “there are no safe treatment options for high cholesterol in childhood” compared to respondents who do not report caring for pediatric patients (Table 1).

Most respondents reported that they would start an individual with HeFH on a statin medication between 18 and 29 years of age, with male patients more likely to be started at a younger age than female patients. Only 17% of respondents would start a male pediatric patient with HeFH on a statin, and only 14% would start a female HeFH patient on a statin. Cardiologists were slightly more likely to report starting statins in pediatric patients than PCPs, although these differences did not reach statistical significance (*p* = 0.12 for male patients and *p* = 0.47 for female patients) (Figure 4).

Multiple logistic regression identified features that were associated with higher or lower likelihood of prescribing a statin for pediatric patients overall, male patients, or female patients (Table 2). No results but one reached statistical significance. PCPs were somewhat less likely than cardiologists to prescribe a statin for a male patient but this was not a factor for overall or female patients. Physicians practicing in the Southwest (compared to Midwest) were less likely to consider prescribing a statin in a male pediatric patient, however the clinical significance of this result is unclear. PCPs were less likely than cardiologists to consider prescribing a statin in a male patient with HeFH under age 18 (OR = 0.79, 95% CI = 0.61–1.02). Physicians who reported previously diagnosing any patient with HeFH were more likely to prescribe a statin in male (OR = 1.34, 95% CI = 0.99–1.81) or female (OR = 1.31, 95% CI = 0.99–1.72) patients under 18 years old.

## 4. Discussion

Our study shows that only 54% of surveyed cardiologists and PCPs would “always” cascade screen a potentially affected first-degree adult relative of an individual with HeFH, which is lower than the number that would monitor LDL-C in someone with “a strong family history of high cholesterol or premature ASCVD”. The majority indicated willingness to screen pediatric patients for HeFH, but very few reported screening at the age recommended by guidelines or willingness to start treatment with a statin at ages recommended by guidelines. 

These findings demonstrate that these physicians surveyed are not recognizing that, as an autosomal dominant disorder, first-degree relatives should be offered screening. This lack of understanding regarding screening recommendations for HeFH is supported by findings of previous studies [21,22,23]. In contrast, more (68%) would begin monitoring LDL-C in young adults with “a strong family history” of premature ASCVD or high cholesterol, but one third of PCPs and 31% of cardiologists would not start monitoring until a patient was at least 30 years old. Ultimately, among the physicians surveyed, knowledge that their patient had HeFH (as opposed to high cholesterol, with a family history of high cholesterol) did not lead them to be more aggressive with cholesterol screening recommendations. This contrast could be due either to a desire to focus care recommendations on the patient alone (as opposed to making care recommendations for the patients’ family members who may not be under their care) or due to a lack of understanding of HeFH inheritance patterns and screening recommendations, or both. This result is consistent with a survey of physicians in the United Kingdom, 50% of whom were not aware that familial hypercholesterolemia has an autosomal dominant pattern of inheritance [24]. 

Agreement to cascade screen for pediatric relatives was different, however. Seventy-four percent of respondents indicated, when asked, that they would be willing to check a cholesterol level in a child or would recommend a child get their cholesterol checked if the child’s parent was their patient and had HeFH. Unsurprisingly, physicians who report caring for pediatric patients were more likely to report that they would screen during childhood. However, very few respondents (7.3% of all surveyed) had sufficient knowledge of pediatric FH recommendations to choose the correct age range during which a pediatric patient with a HeFH parent should be screened (2 years old). Physicians caring for children were slightly more likely to choose the correct age range compared to physicians who did not care for children (8.0% vs. 6.9%, *p* < 0.001). 

Among physicians who would not screen until adulthood, the most common cited reason was lack of familiarity with pediatric cholesterol guidelines. These results further support that physicians are generally unfamiliar with cascade screening recommendations for HeFH. However, physicians who only care for adults are still obligated to understand screening guidelines, so they are able to recommend that the children of their adult patients with HeFH undergo cholesterol screening. The second most common reason cited was “their children are not my patients”, furthering the idea that physicians may be motivated by a desire to focus their care recommendations on their patient alone, and not make recommendations for family members that are not under their direct care. This is consistent with findings from another study [25].

Very concerning is that individuals who report caring for pediatric patients were more likely to agree with the statement “the evidence showing that screening children [for high cholesterol] prevents ASCVD is insufficient”. This may be due in part to conflicting guidelines surrounding cholesterol screening for children. In 2011, the National Heart, Lung, and Blood Institute released pediatric cardiovascular risk reduction guidelines that recommended, among other items, that all children have a non-fasting cholesterol screen performed between 9 and 11 years and again between 17 and 21 years old. This is in addition to standard “selective screening” recommendations for children with high-risk medical conditions or family history of high cholesterol or premature ASCVD [14]. These guidelines were endorsed by the American Academy of Pediatrics. However, in 2016, the United States Preventive Services Task Force updated their pediatric cholesterol screening recommendation, giving it an “I” rating, indicating that the evidence was insufficient to recommend for or against universal cholesterol screening [24]. Of note, this statement did not address selective screening recommendations. These conflicting guidelines may lead to lower rates of cholesterol screening and concern that evidence linking childhood high cholesterol to adult ASCVD is lacking. Given the very strong evidence that cholesterol levels in children and young adults predict the development of atherosclerosis, it is very important that current medical practice takes this into account so that these individuals are screened appropriately. 

Similarly, individuals caring for pediatric patients were more likely to express concern about treatment safety in childhood. This could reflect lack of knowledge about pediatric statin safety data in general or a generalized concern about chronic medication use in children. Conversely, it could represent in-depth knowledge of pediatric statin safety data, which consistently shows safety and efficacy for short- and intermediate-length treatment but is very limited beyond about 20 years of treatment [8]. Physicians who did not report caring for pediatric patients were less likely to express this concern, either because they did not have enough prior knowledge of pediatric treatment to have an opinion regarding the subject, or they thought pediatric statin safety data were adequate. The lack of confidence for treating children and young adults with a statin is concerning given current evidence of the importance of LDL-C lowering as soon as possible given its known safety. 

This study has important limitations. While our study was the largest survey of physicians ever conducted on HeFH and has geographical representation within the US, it is uncertain if survey respondents were representative of all primary care physicians and cardiologists in the US. In particular, the sample of physicians who reported caring for pediatric patients was limited. Respondents were selected if they answered yes to having patients with LDL-C ≥ 190 mg/dL, and therefore, there is a possibility of bias among participants with a specific interest in dyslipidemia, although those who self-identified as lipid specialists were excluded from participation. Information on number of years in practice, sex, or ethnicity of respondents was not available.

## 5. Conclusions

Despite HeFH guidelines recommending cascade screening of potentially affected relatives and starting affected individuals on statin therapy at 8–10 years old, many cardiologists and PCPs do not report routine cascade screening, nor screening for individuals with a strong family history of premature ASCVD or high cholesterol. Those who see pediatric patients are more likely to screen for HeFH but, conversely, have more concerns about the benefit and safety of statin treatment in childhood. Just as concerning, very few indicated willingness to start a pediatric patient with HeFH on a statin. More education is needed among primary care physicians and cardiologists to recognize and treat HeFH and screen family members, particularly pediatric family members. It may also be that education is not enough to change physician behaviors and that innovative ways of actively changing practices are needed. 

## Figures and Tables

**Figure 1 jcm-10-03090-f001:**
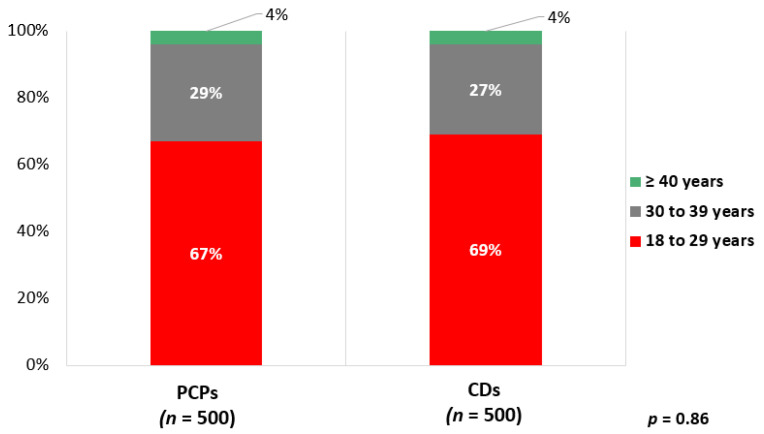
Age at which PCPs and cardiologists report monitoring LDL-C in adults with strong family history of high cholesterol or premature ASCVD. PCPs, primary care physicians; CDs, cardiologists.

**Figure 2 jcm-10-03090-f002:**
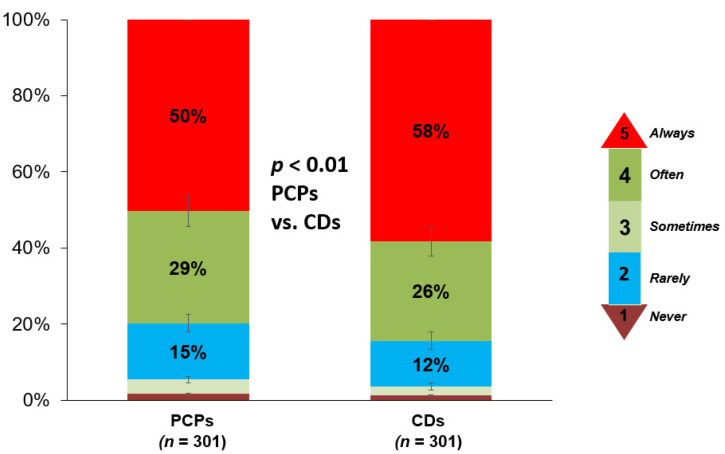
Frequency at which PCPs and cardiologists report recommending cascade screening for a first-degree relative of a patient with HeFH. PCPs, primary care physicians; CDs, cardiologists.

**Figure 3 jcm-10-03090-f003:**
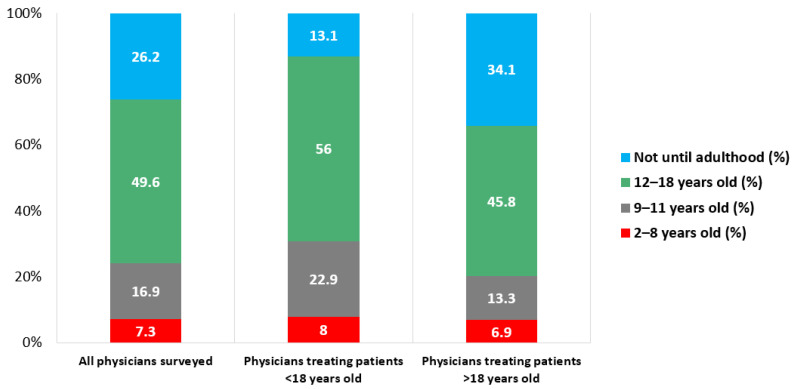
Age at which PCPs and cardiologists recommend cascade screening for pediatric family members of a patient with HeFH. PCPs, primary care physicians; HeFH, heterozygous familial hypercholesterolemia.

**Figure 4 jcm-10-03090-f004:**
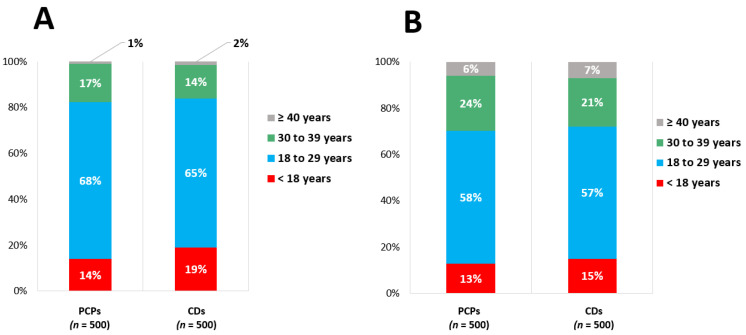
Age at which PCPs and cardiologists advise starting statin therapy in (**A**) male and (**B**) female patients with known HeFH. PCPs, primary care physicians; HeFH, heterozygous familial hypercholesterolemia.

**Table 1 jcm-10-03090-t001:** Reasons provided by 262 respondents for declining to check cholesterol in pediatric patients.

	All Respondents	Respondents Treating Patients < 18 Years	Respondents Only Treating Patients ≥ 18 Years	*p*-Value
*n*	262	49	213	
Not familiar with pediatric guidelines (%)	75.6	57.1	79.8	<0.001
Feel there is insufficient evidence that screening children prevents ASCVD (%)	19.5	42.9	14.1	<0.001
Their children are not my patients (%)	25.2	22.5	25.8	0.62
Feel there are “no safe treatment options” for high cholesterol in childhood (%)	8.4	22.5	5.2	<0.001

ASCVD, atherosclerotic cardiovascular disease.

**Table 2 jcm-10-03090-t002:** Stepwise logistic regression analysis for factors associated with willingness to prescribe a statin for a pediatric patient with HeFH. HeFH, heterozygous familial hypercholesterolemia; PCP, primary care physician; VA, Veterans Affairs; LDL-C, low-density lipoprotein cholesterol.

	Male or Female Patient	Male Patient Only	Female Patient Only
**Characteristic**	OR (95% CI)		
PCP vs. Cardiologist	X	0.79 (0.61–1.02)	X
United States Region	X		X
Northeast vs. Midwest		0.77 (0.52–1.15)	
Pacific vs. Midwest		0.76 (0.50–1.15)	
Southeast vs. Midwest		0.57 (0.38–0.86)	
Southwest vs. Midwest		0.89 (0.58–1.36)	
Practice Location Description		X	
Suburban vs. Rural	1.08 (0.71–1.65)		1.11 (0.73–1.70)
Urban vs. Rural	1.41 (0.91–2.18)		1.46 (0.94–2.26)
Practice Setting VA/Government vs. Academic Health system vs. Academic Private vs. Academic Other vs. Academic	X	X	X
Physician had previously diagnosed a patient with LDL-C ≥ 190 mg/dL as having HeFH	1.27 (0.96–1.68)	1.34 (0.99–1.81)	1.31 (0.99–1.72)

X denotes predictors applied to the stepwise regression model with *p* > 0.15 that were subsequently excluded.

## Data Availability

The data used to support the analysis in this manuscript can be requested from the National Lipid Association.

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
