# Peer review of "Cascade Screening and Treatment Initiation in Young Adults with Heterozygous Familial Hypercholesterolemia"

_jcm, 2021, doi:10.3390/jcm10143090_

Round 1

Reviewer 1 Report

The authors report the result of an interesting survey into knowledge and clinical practice in screening and treating young HeFH patients. This is an important topic since the results of this study and others show that adherence to FH screening guidelines is low. Although limited by obvious reasons for reporting results of a self-reported survey, the study appears to be well performed and the resulting article is well written and easy to read. I have only some minor comments:

  1. “Physicians who reported certification by the American Board of Clinical Lipidology or Accreditation Council for Clinical Lipidology were excluded, as were physicians with a primary specialty of general pediatrics or a pediatric subspecialty.” How does this statement precisely relate to question 6 in the questionnaire? It appears to me that the authors cannot be sure that they excluded members of these boards using this single question.
  2. What is the basis of using a P>0.15 for predictor selection in the model? It appears that this cutoff was chosen to be able to report any results of this endeavor since only one result was significant (Southeast vs. Midwest). Although I understand that this result might be disappointing, table 2 clearly shows that there is no relation between the predictors tested and the likelihood of prescribing a statin to pediatric patients. The only reported significant result (southeast vs. Midwest) is in my view a spurious finding due to multiple testing in this analysis. I would recommend changing this sentence “Very few results reached statistical significance” to “none but one result reached statistical significance”. Moreover, I urge the authors to report why they choose a p-value cutoff of p>0.15 in the method section.
  3. Regarding figure 3: the x-axis and legend are somewhat confusing due to reporting ages in both. Can the authors make the x-axis labels clearer to make the figure understandable without needing to read the main text?

Reviewer 2 Report

Cascade Screening and Treatment Initiation in Young Adults with Heterozygous Familial Hypercholesterolemia is an important contribution to the survey literature.  After specifically excluding pediatricians and physicians with a specialty in clinical lipidology, the authors surveyed 500 PCPs and 500 cardiologists.  The specific aim of the study was to measure physician’s self-reported compliance with recommendations and knowledge regarding current evidence supporting cascade screening of relatives potentially affected by HeFH, and for timing of statin initiation in young adults and children.  The findings of this survey are disappointing but not surprising.  Surveyed physicians are less likely to implement cascade screening in relatives of an individual with known FH than they are individuals with a “strong family history of high cholesterol or premature ASCVD.” While physicians expressed a willingness to preform lipid screening in the child of a parent with known HeFH, only 7.3% of physicians reported that they would screen between the ages of 2-8, in fact most physicians were decades off in terms of when they would embark on both screening and lipid lowering therapies with statins.  

Importantly, this paper nicely highlights the damage done when different trusted entities come to conflicting conclusions regarding lipid screening in children.  While both the NHLBI and the AAP both endorse screening at 2 with a family history of ASCVD or FH and universal screening with a non-fasting lipid panel between the ages of 9-11 and again between ages 17-21, the USPSTF gave such screening an “I” rating, indicating the evidence was insufficient to recommend for or against it.

The authors also provide strong evidence linking statin initiation in HeFH children with a reduction in ASCVD morbidity and mortality in adulthood.

Readers of the Journal of Clinical Medicine, particularly those who treat primarily adults, should be interested to learn about pediatric guidelines.  This article challenges physicians to understand that FH is an autosomal dominant disorder and as such, when caring for a person with FH, the unit is the family, not just the individual patient.

The overall merit of this work is high because it exposes a widely pervasive lack of understanding of current evidenced based guidelines and resulting inadequate timely screening and initiation of statins in children with HeFH.  Additionally,with the USPSTF having just announced that they will re-visit their evaluation of the evidence for/ against pediatric lipid screening, this paper is timely.
